# Two New Potential Therapeutic Approaches in Radiation Cystitis Derived from Mesenchymal Stem Cells: Extracellular Vesicles and Conditioned Medium

**DOI:** 10.3390/biology11070980

**Published:** 2022-06-28

**Authors:** Carole Helissey, Nathalie Guitard, Hélène Théry, Sylvie Goulinet, Philippe Mauduit, Maria Girleanu, Anne-Laure Favier, Michel Drouet, Charles Parnot, Cyrus Chargari, Sophie Cavallero, Sabine François

**Affiliations:** 1Clinical Unit Research, HIA Bégin, 94160 Saint-Mandé, France; 2Department of Radiation Biological Effects, French Armed Forces Biomedical Research Institute, 91220 Brétigny-sur-Orge, France; nathalie.guitard@intradef.gouv.fr (N.G.); helene.thery@intradef.gouv.fr (H.T.); michel1.drouet@intradef.gouv.fr (M.D.); cyrus.chargari@gustaveroussy.fr (C.C.); sophie.cavallero@intradef.gouv.fr (S.C.); sabine.francois@intradef.gouv.fr (S.F.); 3Technical Platform iVEs, UMRS-MD 1197 (INSERM), Paul Brousse Hospital, 94800 Villejuif, France; sylvie.goulinet@inserm.fr (S.G.); philippe.mauduit@inserm.fr (P.M.); 4Imaging Unit, Department of Platforms and Technological Research, French Armed Forces Biomedical Research Institute, 91220 Brétigny-sur-Orge, France; maria.girleanu@intradef.gouv.fr (M.G.); anne-laure.favier@intradef.gouv.fr (A.-L.F.); 5Cureety, 33 rue de l’Amirauté, 22100 Dinan, France; charles@cureety.com; 6Department of Radiation Oncology, Gustave Roussy Comprehensive Cancer Center, 94800 Villejuif, France; 7INSERM U1030, Université Paris Saclay, 94043 Le Kremlin Bicêtre, France

**Keywords:** mesenchymal stem/stromal cells, conditioned medium, extracellular vesicles, human bladder fibroblasts, radiation cystitis

## Abstract

**Simple Summary:**

Pelvic radiotherapy is a major therapeutic weapon in the management of pelvic tumors. Despite improvements in radiation techniques, pelvic irradiation is responsible for symptoms that impact the quality of life of our patients. Thus, the incidence of radiation cystitis remains stable over time and the therapeutic possibilities remain limited. Extracellular vesicles (MSC-EVs) or conditioned medium from human mesenchymal stromal cells (MSC-CM) have demonstrated therapeutic potential by promoting tissue repair. We have evaluated their efficacy in preventing fibrosis in a model of radiation cystitis *in vitro*.

**Abstract:**

**Background:** Radiation cystitis (RC) results from chronic inflammation, fibrosis, and vascular damage. The urinary symptoms it causes have a serious impact on patients’ quality of life. Despite the improvement in irradiation techniques, the incidence of radiation cystitis remains stable over time, and the therapeutic possibilities remain limited. Mesenchymal stem/stromal cells (MSC) appear to offer^2^ a promising therapeutic approach by promoting tissue repair through their paracrine action via extracellular vesicles (MSC-EVs) or conditioned medium from human mesenchymal stromal cells (MSC-CM). We assess the therapeutic potential of MSC-EVs or MSC-CM in an *in vitro* model of RC. **Methods:**
*in vitro* RC was induced by irradiation of human bladder fibroblasts (HUBF) with the small-animal radiation research platform (SARRP). HUBF were induced towards an RC phenotype after 3 × 3.5 Gy irradiation in the presence of either MSC-EVs or MSC-CM, to assess their effect on fibrosis, angiogenesis, and inflammatory markers. **Results:** Our data revealed *in vitro* a higher therapeutic potential of MSC-EVs and MSC-CM in prevention of RC. This was confirmed by down-regulation of α-SMA and CTGF transcription, and the induction of the secretion of anti-fibrotic cytokines, such as IFNγ, IL10 and IL27 and the decrease in the secretion of pro-fibrotic cytokines, IGFBP2, IL1β, IL6, IL18, PDGF, TNFα, and HGF, by irradiated HUBFs, conditioned with MSC-EVs or MSC-CM. The secretome of MSC (MSC-CM) or its subsecretome (MSC-EVs) are proangiogenic, with the ability to induce vessels from HUVEC cells, ensuring the management of bladder vascular lesions induced by irradiation. **Conclusion:** MSC-EVs and MSC-CM appear to have promising therapeutic potential in the prevention of RC *in vitro*, by targeting the three main stages of RC: fibrosis, inflammation and vascular damage.

## 1. Introduction

External pelvic radiation therapy is an important tool in the treatment modality for pelvic cancers such as prostate cancer. Irradiation techniques have improved over time, such as intensity-modulated radiation therapy (IMRT), stereotactic radiation therapy, and image-guided brachytherapy. These techniques allow us to deliver increasingly effective doses in smaller volumes while significantly improving treatment tolerance [1].

However, the bladder is a critical organ sensitive to low-dose radiation. Despite improvements in technology, pelvic irradiation remains a cause of acute and/or late adverse events affecting the bladder. The term “radiation cystitis” includes all damage and symptoms of the bladder following radiation to the pelvic organs.

Severity is related to radiation exposure, total dose administered, and schedule of administration and fractionation. Patients with hypertension, diabetes mellitus, history of abdominal surgery, and concurrent chemotherapy have been reported to have a higher risk of radiation cystitis, especially in advanced patients [2]. Manea et al. suggested that bladder neck, after high dose exposures (such as after brachytherapy treatment) may be at higher risk of [3].

Acute radiation cystitis was defined as any adverse event that occurred during or within three months of the end of radiation therapy. Clinical symptoms may include pollakiuria, cystalgia, increased frequency and frequency of urination during the day and night (polyuria), dysuria, and increased urinary urgency. Late radiation cystitis is defined as a pelvic radiation-related adverse event that occurs at least three months and possibly years after completion of radiation therapy. The most typical clinical feature is repeated hematuria of varying severity.These adverse events can affect the patient’s quality of life. The clinical management of storage symptoms for acute and late radiation cystitis is largely symptomatic with analgesics, anti-inflammatory drugs, intravesical instillations, hyperbaric oxygen therapy (HBOT) and can go up to cystectomy and urinary diversion. Good hydration is recommended for patients in order to increase diuresis, cleanse the bladder, and avoid urinary obstruction resulting from blood clots [4].

RC damage includes vascular lesions, fibrosis and inflammation. Human bladder fibroblasts (HUBFs) are the key cell in this adverse event.

Mesenchymal stem cells/stromal cells (MSC) are a promising alternative therapy for the treatment of fibrosis due to their proangiogenic potential, immunomodulatory effects, and promotion of tissue repair [5,6]. However, MSC present many challenges, such as their variability, scalability and deployment, limit their clinical applications [7]. 

The beneficial effects of MSCs can be attributed to their paracrine effects. This can be achieved by regulating its environment by secreting specific components (MSC-CM) and including extracellular vesicles (MSC-EV) [8].

The beneficial effects of MSCs may be attributed to their paracrine action. This may be achieved through the conditioning of its environment by secretion of specific components (MSC-CM) and among them extracellular vesicles (MSC-EVs) [8].

In fact, MSC-EVs, including exosomes and microvesicles, have shown remarkable therapeutic properties in many pathophysiological conditions, with the potential to repair various damaged or diseased tissues and organs, such as nervous system, heart, lung, kidney, bone marrow or cartilage, among others [9,10]. Likewise, MSC-CM has shown therapeutic potential in various fields such as myocardial infarction, stroke and acute and chronic ischemia, neurodegenerative diseases, spinal cord injury, alopecia, acute and chronic wounds, acute liver injury and failure, lung injury, periodontal tissue injury, male infertility and soft tissue and bone defects [11,12].

Thus, MSC-EVs and MSC-CM appear to be promising therapeutic candidates to slow the development of radiation fibrosis and maintain tissue integrity in the long term.

To date, few studies have evaluated their antifibrotic, proangiogenic and immunomodulatory potential, the three key steps in the management of radiation cystitis.

In our study, we assess the therapeutic potential of MSC-EVs and MSC-CM in a human *in vitro* model of RC.

We conducted a comparative study on the ability of MSC-EVs and MSC-CM in the regulation of gene transcription, modulation of the secretion of pro- and anti-angiogenic proteins and the secretion of pro- and anti-inflammatory cytokines in human bladder fibroblasts (HUBFs).

Our results demonstrated the therapeutic potential of MSC-EVs and MSC-CM in the prevention of RC, as confirmed by down-regulation of α-SMA, CTGF and Col1α2 transcription and up-regulation of antifibrotic genes, such as MMP2. An increase was observed in secretion of pro-angiogenic factors including IGFBP2, IL1β, IL6, IL18, HGF and TNF-α by MSC-EVs and MSC-CM, and anti-fibrotic cytokines such as IFN-γ, IL10, IL27 by conditioned HUBFs.

## 2. Materials and Methods

### 2.1. Mesenchymal Stem/Stromal Cell (MSC) Culture and Characterization

Human mesenchymal stem/stromal cells (MSCs) were isolated from bone marrow aspirates, obtained from Lonza (Walkersville, MD, USA, WV IM-105), sourced from healthy (BMI < 25, non-diabetic) informed consent donors aged 30–40 years. Ten millimeters of unprocessed human bone marrow contained an average of 20 × 10^6^ nucleated cells/mL. Human cells were seeded at a density of 0.16 × 10^6^ cell/cm^2^ in mesenchymal stem cell growth medium (MEM Alpha Medium, Gibco ref. 22561-021) containing Amphotericin B 250 μg/mL (ref. Gibco041-95780), 1% L-glutamine 200 mM 100× (ref. Gibco 25030-024), 1% Penicillin-Streptomycin 10,000 U/mL, 100×, (ref. Gibco15140122) and fetal bovine serum 10% (Hyclone research grade fetal bovine serum, collected in South America, GE Healthcare Life sciences, ref. SV30160.03). Cultures were kept at 37 °C, 5% CO_2_, and 95% humidity. Twenty-four hours later, non-adherent cells were withdrawn, and refreshment with mesenchymal stem cell growth medium was carried out. Culture medium was changed 3 times per week, until 80% confluence was obtained. At this stage, the cells are said to be at passage 0. At each phase of expansion, MSCs were seeded at a density of 5000 cells per cm^2^. During this study, the cells were amplified up to passage 3 (P3), to limit any phenotypic change in the primary cells. At P3, their differentiation potential and phenotype were analyzed following International Society for Cell and Gene Therapy (ISCT) recommendations [13,14]. For phenotyping, the cells were labeled with the CD90-FITC, CD105-FITC, CD73-PE, CD34-PE and CD45-PE antibodies (BD Biosciences) and analyzed by flow cytometry. For differentiation potential, MSCs were cultured with lineage-specific media, StemPro Adipogenesis Differentiation Kit (Gibco, A10070-01), StemPro Osteogenesis Differentiation Kit (Gibco, A10072-01) StemPro Chondrogenesis Differentiation Kit (Gibco, A10071-01) for 21 days to induce cell differentiation. Differentiation into adipocytes, osteocytes and chondrocytes was detected by Oil Red O, alizarin red and alcian blue, respectively, and observed upon visible light microscopy.

### 2.2. Isolation of MSC-EVs et MSC-CM Collection

At passage 3 of MSCs, (80% of confluence) the cells were washed at least three times with sterile 1× PBS (Gibco) and re-cultured in serum-free growth medium for 72 h. Then, the medium was collected and centrifuged (3000× *g*, 10 min) to remove whole cells and debris. A part of the supernatant was kept at −80 °C (MSC-CM), and the other part was filtered by tangential flow filtration (TFF) using syringes with single-use hollow fiber filters (Repligen, mPES 500k) to isolate and concentrate MSC-EVs (Figure 1). Quantification and sizing of particles present in MSC-CM and MSC-EVs samples was carried out using the NanoSight NS300 instrument (Malvern Panalytical). CD63 levels, widely used as EVs markers, were measured, in MSC-CM and MSC-EVs, using the ExoELISA-ULTRA CD63 kit (System Biosciences, Palo Alto, CA, USA) according to the manufacturer’s protocol. EVs were visualized by cryo-electron microscopy (Cryo-EM). For the sample preparation, Quantifoil grids R1.2/1.3 from Agar Scientific were glow-discharged (20 s, 5 mA) using a Leica EM ACE600 system. Then, 2.5 µL of vesicles in PBS (1.4 × 10^9^ EV/mL) was deposited on the carbon side of the EM grid, blotted for 1s with a blot force of 2 and plunge-frozen in liquid ethane using a Vitrobot Mark IV (ThermoFisher, Waltham, MA, USA). For the cryo-EM analysis, a Titan Krios 300KV cryo-electron microscope was used, equipped with a Falcon III direct electron detector (FEI, Hillsboro, OR, USA) and CMOS CETA detector (FEI, Hillsboro, OR, USA). The images were taken on the CETA detector at a magnification of 37,000× with a pixel size of 0.23 nm, a defocus of −2 µm using an aperture of 100 µm. The exposure time of the images was 1 s with an accumulated total dose of 40 e-/Å^2^ per image. No filtering procedures were applied to images.

### 2.3. Analysis of the Angiogenic Potential of MSC-EVs and MSC-CM

Protein contents of MSC-EVs and MCS-CM samples were measured using a BCA protein assay kit (Thermo Scientific Pierce, Rockford, IL, USA). The expression of angiogenic protein was analyzed using Proteome Profiler Human Angiogenesis Array (R&D Systems, catalogue number ARY007). Briefly, MCS-EVs and MCS-CM were mixed with the biotinylated detection antibody cocktail and incubated with nitrocellulose membranes containing spotted specific antibodies. Following adsorption of the biotinylated-antibody–antigen complexes, membrane was washed and further incubated with Streptavidin-HRP for quantification. Cytokine detection was performed using a ChemiDoc XRS+ System (BioRad, California, CA, USA). The signal density of each blot was quantified by dot blot analysis on NIH ImageJ analysis software 1.8.0. The level of 3 pro-angiogenic proteins was measured with a Quantikine ELISA kit (R&D systems): IL8 (D8000C), MCP-1 (DCP00) and CXCL16 (DCX160), following the manufacturer’s instructions.

#### 2.3.1. *In Vitro* Tube Formation Assay

Geltrex LDEV-Free Reduced Growth Factor Basement Membrane Matrix (Gibco, 100 µL/well) was coated onto 24-well plates and cultured in a 37 °C for 30 min for matrix gel polymerization. Human umbilical vein endothelial cells (HUVEC, Gibco) were removed from culture after 7 days in LSGS-supplemented Medium 200 (Gibco), trypsinized and resuspended in LSGS-supplemented Medium 200. HUVECs (5 × 10^4^ cell/well) were seeded into each well in different experimental conditions: LSGS-supplemented Medium 200 (positive control) or LSGS-supplemented Medium 200 with MSC-EV (2.5 × 10^7^ VE/well) or LSGS-supplemented Medium 200 with MSC-CM (1.8 × 10^7^ VE/well). After 12 h incubation, the cells were stained with 2 μg/mL of Calcein, AM (Invitrogen, Waltham, MA, USA) incubated for 30 min at 37 °C. Fluorescence images were captured at a 10× magnification using CKX53 brightfield and epifluorescence microscope (Olympus, Allentown, PA, USA). Number of nodes, number of junctions and segment length were counted from the collected images using the NIH ImageJ analysis software, to quantify the angiogenic network formation.

#### 2.3.2. *In Vitro* Model of Radiation-Induced Bladder Myofibroblasts

Normal human bladder fibroblasts (HUBFs) were isolated from the normal human urinary bladder, (#FC-0050, CellSystems, Troisdorf, Germany), plated at 2500 cells/cm^2^ and expanded in complete medium for fibroblasts (FibroLife basal Medium associated with FibroLife S2 LifeFactors Kit (ref. cellSystems #LL-0011)). Cultures were kept at 37 °C, 5% CO_2_, and 95% humidity. HUBFs were expanded and irradiated *in vitro* (220 kV, 0.97 Gy/min) using a fractionation scheme of 3 × 3.5 Gy at a 24 h interval (SARRP—small-animal radiation research platform). The irradiation protocol was based on the work of Andreassen et al. on fibroblasts isolated from healthy human skin biopsies [15] (Figure 2).

### 2.4. HUBFs Preconditioned with MSC-EVs or MSC-CM

Seventy-two hours before irradiation, the HUBFs were inoculated in 6-well plates at a density of 50,000 cells per well. They were incubated at 37 °C either in 2 mL complete medium for fibroblasts alone, or in 2 mL complete medium for fibroblasts containing 1.45 × 10^8^ MSC-EVs/mL, or in 2 mL of complete medium for fibroblasts containing 0.25 × 10^8^ MSC-CM/mL (Figure 2).

### 2.5. RNA Extraction and Reverse Transcription, Quantitative Real-Time Polymerase Chain Reaction to Analyze Fibrosis Genes

From D2 to D7, the culture supernatants and the cell pellets were harvested and stored at −80 °C. RNA was extracted from these pellets using NucleoSpin^®^ RNA Set for NucleoZOL (Macherey-Nagel, ref 740406, Hoerdt, France). Briefly, after lysing the sample in NucleoZOL and pelleting contaminating DNA and proteins, the supernatant was mixed with an optimized binding buffer for efficient binding of small and large RNA to the silica membrane. RNA was quantified using a nanodrop. Reverse transcription of 1 μg total RNA per reaction was carried out using iScript cDNA Synthesis Kit (BIO RAD RNA) according to the manufacturer’s specifications (Figure 2).

TaqMan gene expression assays (Life Technologies; Applied Biosystems, Waltham, MA, USA) were used to quantify the following transcripts: α-SMA (Hs00426835_g1), CTGF (Hs00170014_m1), TIMP1 (Hs01092512_g1), Col3α1, (Hs00943809_m1), COL1α2 (Hs01028956_m1), MMP2 (Hs01548727_m1), TGF-β1 (Hs00998133_m1. Gene expression levels were normalized to GADPH (Hs02786624_g1), which was used as reference gene for HUBF cells. Corresponding to mRNA, cDNA was used at a concentration of 25 ng/mL. qRT-PCR was performed with cDNA from 50 ng total RNA and TaqMan Fast Universal Master Mix (Applied Biosystems, Waltham, MA, USA) on a StepOnePlus instrument (Applied Biosystems, Waltham, MA, USA) under the following standard conditions: 95 °C for 20 s, followed by 40 cycles of 95 °C for 1 s and 60 °C for 20 s. The relative gene expression was calculated via the comparative Ct method as previously described by K. Livak (Applied Biosystems User Bulletin #2, 2001, Waltham, MA, USA). Ct values were normalized to GADPH and used to calculate the relative gene expression using the 2^−ΔΔCt^ method. The fold change was calculated relative to the gene expression of non-irradiated (NIR) HUBFs.

### 2.6. Analysis of HUBF Secretome Cytokine Profile

The cytokine profile was analyzed using Proteome Profiler Human XL Cytokine Array (R&D Systems Catalog Number ARY022B). Briefly, the HUBF secretome was mixed with the biotinylated detection antibody cocktail and incubated with nitrocellulose membranes. The washed membranes were incubated with Streptavidin-HRP. Cytokine detection was performed using a ChemiDoc XRSn+ System (BioRad, Hercules, CA, USA). The signal density of each blot was quantified by dot blot analysis on the NIH ImageJ analysis software.

### 2.7. Statistical Analysis

To determine the effect of radiation exposure, MSC-EVs and MSC-CM on the phenotype and activation of HUBFs into myofibroblasts, variations in gene expression were compared. Statistical significance was calculated using the Mann–Whitney test. Significance for all analyses was set at *p*-value < 0.05 (*), *p* value < 0.01 (**) and *p* value < 0.001 (***). We used GraphPad Prism 7.0 software (https://graphpad-prism.software.informer.com/7.0/). All values were expressed as the Mean and SEM. Each condition consisted of 6–8 samples of HUBFs.

The mRNA level of GAPDH gene was considered as normalizing control gene. We interpreted the fold changes as follows:

If there is a doubling (fold change = 2, Log2FC = 1) between IR and NIR, then IR is twice as large as NIR (or IR is 200% of NIR).

If there is a two-fold decrease (factor change = 0.5, Log2FC = −1) between IR and NIR, then IR is half as large as NIR or IR is 50% of NIR.

In this study, we consider a significant overexpression from 1.5 FC and underexpression from 0.5 FC of IR versus NIR.

## 3. Results

### 3.1. Mesenchymal Stem Cell Characterization

Phenotypic analysis showed that human MSCs used in these experiments were strongly positive for the specific surface antigens CD105, CD73 and CD90 (73.7% ± 3.1%, 72.0% ± 0.6%, 71.40% ± 0.7, respectively). Almost no contamination (0.51% ± 0.1% CD 45^+^ cells and 0.78% ± 0.1% CD 34^+^) by hematopoietic cells was evidenced in the samples (Figure 3A). *in vitro* culture experiments showed that the cells were still able to differentiate into the osteoblast, chondrocyte, and adipocyte lineages (Figure 3B).

### 3.2. EVs and MCS-CM Characterization

The size and concentration of extracellular vesicles in samples were measured by Nanoparticle Tracking Analysis (NTA). NTA demonstrated a comparable particle size distribution between MSC-CM and MSC-EV samples, of 114 and 121 nm, respectively (mode, *n* = 5). These results are confirmed by the observations of vesicle morphology made by means of cryo-electron microscopy. The mean concentration of particles measured by NTA was 2.63 × 10^8^/mL in MSC-CM samples and 1.45 × 10^9^ in MSC-EV samples. This result shows that tangential filtration concentrates the vesicles because they are more numerous in MSC-EVs. Tangential filtration significantly increases by a factor of 5.5 (*p* = 0.0159). The level of CD63, a specific marker for EVs, proves that EVs are present in both MSC-CM and MSC-EVs preparations and that they are more concentrated in MSC-EVs (Figure 4).

### 3.3. In Vitro Human Radiation Cystitis Model

We analyzed this fibrogenic differentiation of HUBFs after irradiation by SARRP, and reported mRNA expression profiles of myofibroblast-associated genes in HUBFs irradiated at day (D) 5 post irradiation compared to non-irradiated HUBF (Figure 5). A significant increase in the gene expression of the pro-fibrosating genes at D5 was observed. The level of myofibroblast marker was significantly increased in irradiated HUBFs. Expression of α-SMA, CTGF, TGF-β1, TIMP1, TIMP2, Col3α1, Col1α2 and matrix metallopeptidases (MMPs) were upregulated in irradiated HUBFs compared to non-irradiated HUBF. At D5 post irradiation, a significant increase was observed in α-SMA (+7.13 FC ±0.96) CTGF (+10.96 FC ±0.19), Col3α1 (+57.23 FC ±1.85) and Col1α2 (+21.62 FC ± 3.26), TIMP1 (+2.09 FC ± 0.14) and MMP2 (+5.02 FC ±0.74), TGF-β (+2.41 FC ± 0.25) compared to non-irradiated HUBFs (Figure 5). These results support the idea that the irradiation protocol leads, at D5 post irradiation, to an activation of HUBFs into myofibroblasts, which are responsible for the development of fibrosis.

### 3.4. Secretome Analysis of Irradiated HUBF Cells

The protein expression of the secretome of irradiated and non-irradiated HUBFs was analyzed using the Proteome Profiler XL cytokine Array Kit (Figure 6). At D5 post irradiation, this semi-quantitative analysis shows a significant increase in pro-fibrotic proteins, IGFBP2, IL1β, IL6, IL18, and TNFα and HGF, within the secretome of irradiated HUBFs compared to non-irradiated HUFBs. At the same time, anti-fibrotic proteins IFNγ, IL10 and IL27 decrease significantly compared to non-irradiated fibroblasts. The HUBFs gene profile at D5 post irradiation with this molecular signature of fibrotic tissue indicates that the radiation-induced bladder myofibroblasts are in a pro-fibrotic environment. This *in vitro* model aims to represent radiation-induced myofibroblasts and their autocrine and paracrine environment at the stage of radiation cystitis, in order to examine cell-based responses, therapeutic responses of MSC-EVs and MCS-CM after radiation exposure.

### 3.5. MSC-EVs and MCS-CM Dowregulate the Expression of Pro-Fibrotic Genes and Up-Regulate Anti-Fibrotic Genes in Irradiated HUBF Cells

MSC-EVs and MCS-CM changed the phenotype of irradiated HUBFs. Indeed, α-SMA gene expression in irradiated HUBFs preconditioned with MSC-EVs and MCS-CM was 3.85 FC ± 0.43 and 5.86 FC ± 0.58, respectively, compared to non-irradiated HUBFs. These expression levels are largely reduced compared to irradiated fibroblasts (+7.13 FC ± 0.96). In particular, when the fibroblasts are preconditioned with MSC-EVs, a significant decrease by a factor of 2 (*p* = 0.0041) is observed compared to irradiated HUBFs (Figure 7).

Likewise, CTGF gene expression of irradiated fibroblasts preconditioned with MSC-EVs and MCS-CM was 2.77 FC ± 0.16 and 0.75 FC ± 0.10, respectively, compared to non-Irradiated HUBFs. These expression levels are notably reduced for the two preconditionings compared to irradiated HUBFs (10.96 FC ± 0.19). The expression level of irradiated HUBFs preconditioned with MSC-EVs is significantly reduced compared to irradiated HUBFs by a factor of 4 (*p* < 0.001). Additionally, the expression level of irradiated HUBFs preconditioned with MCS-CM is significantly reduced by a factor of 15 (*p* < 0.001) compared to irradiated HUBFs. Preconditioning with MCS-CM results in a significant decrease in CTGF gene expression by a factor of 4 (*p* < 0.001) compared to irradiated HUBFs preconditioned MSC-EVs (Figure 7).

Next, the effect of MSC-EVs and MCS-CM on the gene expression of extracellular matrix components such as type I (Col1α2) and type III (Col3α1) collagen was analyzed (Figure 7).

Col1α2 gene expression of irradiated HUBFs preconditioned with MSC-EVs and MSC-CM was 15.39 ± 4.54 and 8.01 ± 1.36, respectively, compared to non-irradiated HUBFs. This expression level is significantly reduced with MC-MSC preconditioning by a factor of 2.7 (*p* < 0.001) compared to irradiated HUBFs (21.62 FC ± 3.26). No significant difference was found between the two preconditionings (Figure 7).

MMP2 gene expression of irradiated HUBFs preconditioned with MSC-EVs and MSC-CM was 6.17 ± 0.47 and 7.86 ± 1.41, respectively, compared to non-irradiated HUBFs. The increase in these expression levels is not significant compared to irradiated fibroblasts (5.02 FC ± 0.74) due to a large degree of variability (Figure 7).

These results seem to suggest an almost equivalent anti-fibrotic potential of MSC-EVs and MSC-CM. MSC-EVs and MSC-CM also take part in the wound-healing process after irradiation while inhibiting fibrosis by maintaining production of Col3α1, TIMP1, and TGF-β1. Pre-conditioning before irradiation did not lead to variation in Col3α1 gene expression by irradiated fibroblasts. Col3α1 gene expression of the HUBFs preconditioned with MSC-EVs and MSC-CM was 75.35 ± 12.43 and 77.60 ± 9.51, respectively, compared to non-irradiated HUBFs (+57.23 FC ± 1.85). This increase in the expression level is not significant compared to irradiated HUBFs (Figure 7). Overexpression of Col3α1 by non-irradiated HUBFs, in order to maintain homeostasis was notable. Tissue inhibitors of metalloproteinases (TIMPs) are tissue specific, endogenous inhibitors of metalloproteinases, including matrix metalloproteinases (MMPs). The TIMP1 gene expression of irradiated HUBFs preconditioned with MSC-EVs and MSC-CM was 3.22 ± 0.18 and 6.86 ± 1.20, respectively, compared to non-irradiated HUBFs. The expression level of TIMP1 is significantly increased by the two preconditionings compared to irradiated HUBFs (+2.09 FC ± 0.14). Preconditioning with MSC-CM appears to be significantly greater by a factor of 2 (*p* = 0.0226) compared to those preconditioned with MSC-EVs (Figure 7). TGF-β1 gene expression of irradiated HUBFs preconditioned with MSC-EVs and MSC-CM was 4.23 ± 0.59 and 7.51 ± 0.94, respectively, compared to non-irradiated HUBFs. The expression level of TGF-β1 was significantly increased for the two preconditionings compared to irradiated HUBFs (2.41 FC ± 0.25). This elevation of TGFβ1 gene expression of HUBFs preconditioned with MSC-CM appears to be significantly greater by a factor of 2 (*p* = 0.0218) compared to HUBFs preconditioned by MSC-EVs before irradiation (Figure 7). These results underline that MSC-EVs and MSC-CM also contribute to the wound-healing process following irradiation while inhibiting fibrosis by maintaining Col3α1, TIMP1 and TGF-β1 production.

### 3.6. MSC-EVs and MSC-CM Modulate the Secretome of Irradiated HUBFs

The protein expression of the secretome in irradiated HUBFs pre-conditioned with MSC-EVs and MSC-CM was analyzed using the Proteome Profiler Human XL cytokine Array Kit and compared to the protein expression of the secretome in irradiated HUBFs (Figure 8). At D5 post irradiation, this semi-quantitative analysis shows a significant decrease in pro-fibrotic proteins within the secretome of irradiated HuFBs pre-conditioned with MSC-CM and MSC-EVs compared to irradiated HUBFs: IGFBP2, IL1-β, IL6, IL11, IL18, PDGF, TNFα, HGF in MSC-CM, MSC-EVs and irradiated HUBF groups, respectively. Furthermore, a significant increase was observed in anti-fibrotic proteins within the secretome of irradiated HUBFs preconditioned with MSC-CM and MSC-EVs compared to irradiated HUBFs: with IFNγ, IL10 and IL27 in MSC-CM, MSC-EVs and irradiated HuFB groups, respectively.

This points to an anti-fibrotic potential for the two preconditionings, MSC-EVs and MSC-CM, without evidence of a stronger potential in one or the other.

### 3.7. Potential Pro-Angiogenic of MSC-EVs and MSC-CM

The expression of proteins involved in angiogenesis was analyzed for MSC-EVs and MSC-CM using the Proteome Profiler Human Angiogenesis Array Kit (Figure 9A). This semi-quantitative analysis shows an overexpression of pro-angiogenic proteins by MSC-EV and MSC-CM, such as VEGF, IGFβ2 and IGFβ3 (Figure 9B). Quantification by ELISA also shows overexpression of three proteins essential to angiogenesis, IL-8, MCP1, and CXCL16 (Figure 9C). These data confirm the proangiogenic potential of MSC-EVs and MSC-CM. MSC-EVs seem to express these proteins in greater quantity than MSC-CM.

In order to evaluate the angiogenic potential of MSC-CM and MSC-EVs, endothelial tube formation assays were performed. For the quantification using the NIH ImageJ analysis software, we used three indicators to determine the angiogenic effects of MSC-CM and MSC-EVs: the number of nodes, the number of junctions, and the segments length. As illustrated in Figure 10, HUVECs from incubation with MSC-CM and with MSC-EVs formed more nodes, junctions, and longer segments than the cells incubated with medium only.

This greater proangiogenic potential is what most prominently distinguishes MSC-CM from MSC-EVs in our study.

## 4. Discussion

To the best of our knowledge, our study is the first to develop a model of radio-induced bladder myofibroblasts in a pro-fibrotic environment *in vitro*. We have also demonstrated that MSC-EVs and MSC-CM are two potential candidates in the prevention of radiation cystitis, due to their capacity to inhibit the bladder myofibroblast phenotype, their capacity to decrease the secretion of pro-inflammatory cytokines of HUBFs and, finally, their pro-angiogenic capacities to help maintain vascular integrity and limit tissue ischemia.

Acute radiation tissue injury to the bladder is characterized by edema, hyperemia and inflammation of the mucous membrane. Urothelium is fragile and vulnerable to trauma and infections [16]. In late radiation tissue injury, the submucosal vascularity is damaged by fibrosis of the vascular intima resulting in vessel obliteration and submucosal/muscular fibrosis. This is followed by urothelial atrophy, hypoxia with hypovascularization and ischemia of the bladder leading to the development of fibrosis and atrophy of the bladder tissue [1,17]. Clinical symptoms may include increased urinary urgency and frequency (pollakiuria), both during the day and at night, dysuria, but also cystalgia with bladder spasms, and hematuria. These adverse events have a serious impact on patients’ quality of life and can be life-threatening in some cases. Nowadays, their management is limited to symptomatic treatment and remains a therapeutic challenge.

To date, the pathophysiology of RC remains unclear. However, it has been established that the bladder fibroblast plays a key role in the process of fibrosis, by its role in the overproduction of the extracellular matrix but also by its interaction with the immune system and endothelial cells [18].

Our work initially involved establishing a model of RC *in vitro* (bladder myofibroblasts activated in a radio-induced pro-fibrotic environment) so as to assess the effect of irradiation on the fibroblast and its implication in these three key stages of post-radiation bladder fibrosis.

In our study, a 3.5 Gy dose of radiation per day for 3 consecutive days using SARRP was applied to create an *in vitro* model of radiation cystitis. Our data demonstrated significantly increased mRNA expression of α-SMA, CTGF, Col1α2, Col3α1, TGF-β at D5. Indeed, CTGF had previously been reported to modulate many signaling pathways responsible for tissue remodeling and fibrosis development [19]. CTGF protein induces gene expression and levels of proteins involved in the synthesis of the extracellular matrix (ECM) [20].

TGF-β is the primary factor that drives fibrosis [21]. Activation of TGF-β signaling, for example by fibroblast-specific overexpression of constitutively active TGF-β receptor type 1 (TGF-β-RI), is sufficient to induce a systemic fibrotic disease with progressive fibrosis in multiple tissues [22]. TGF-β has a close association with Matrix MetalloProteinases (MMPs), a family of proteolytic enzymes involved in the degradation and remodeling of extracellular matrix proteins.

Tissue inhibitors of metalloproteinases 1 (TIMP1) expression is increased within these profibrotic environments, which suggests a role for TIMP1 in restricting ECM proteolysis [19,23]. McLennan et al. reported a close interaction between CTGF and TGF-β and TIMP1 in the development of fibrosis in the context of hyperglycemia. However, above all, maintaining a high level of CTGF is necessary to sustain the process of fibrosis [20]. CTGF is therefore one of the essential therapeutic targets in the prevention of fibrosis [24]. Additionally, maintaining the correct level of TGF-β ensures healing and homeostasis while avoiding autoimmune pathologies [18,25,26].

An excess in collagen production is necessary for the development of pulmonary fibrosis. In the pulmonary fibrosis model, less-elastic type-I collagen was found to predominate, perhaps even exclusively [27,28].

These pathological features were consistent and provided a basis for the successful establishment of an *in vitro* radiation cystitis model. In addition, analysis of irradiated fibroblasts secretome at D5 revealed a molecular signature of fibrotic tissue, with increased secretion of pro-fibrotic cytokines, such as IGFBP2, IGFBP3, IL1b, IL4, IL6, IL 11, IL18, IL33, PDGF and TNFα, and a decrease in anti-fibrotic molecules, such as IL10.

IL-1β and TNFα are pro-inflammatory cytokines that are expressed in common chronic pathologies such as rheumatoid arthritis, diabetes, myocardial infarction, and inflammatory lung diseases [29]. IL-10 is a potent anti-fibrotic and immunomodulatory cytokine secreted in response to elevated levels of inflammatory cytokines such as IL-β, TNFα, and interferon (IFN) [30,31].

This modification of the secretome of irradiated HUBF on line the bystander effects of ionizing radiation (the non-targeting effects irradiation), and thus promotes the fibrosis process. This bystander effect has been reported in other *in vitro* models, notably in the process of angiogenesis [32].

Other molecules are also involved in the fibrosis process such as MMP9 and TIMP2. Indeed, Mohammed and Said reported that MMP-9 is constitutively expressed in the intestinal mucosa, and is highly expressed in rat intestines after ϒ-radiation exposure, and its inhibition leads to a tendency to inhibit fibrosis [33]. Wang et al. confirm the role of MMP9 in the process of liver fibrosis, but its role is dynamic over time [34]. Additionally, the different properties of MMP9 must continue to be explored.

In our model, we did not find any significant evolution of MMP9 and TIMP2 gene expression between irradiated and non-irradiated HUBF. Zwaans et al. report no difference in PAI, TIMP1 and TIMP2 expression in urine between prostate cancer patients who underwent pelvic irradiation. Similarly, there was no significant difference in the expression of PAI, TIMP1 and TIMP2 in long-surviving patients with CR symptoms compared to patients who also received pelvic radiotherapy but did not show CR symptoms [35]. Thus, we need further investigation to determine the place of these molecules in the radiation cystitis model.

This *in vitro* model allows for an initial assessment of new anti-fibrotic therapies before studying their effectiveness in vivo, in accordance with the “3R” principles of bioethics: Reduce, Refine and Replace.

The use of MSCs in tissue remodeling and repair has grown notably in recent years. Few studies have looked into the place of MSCs in the management of radiation cystitis. Wiafe et al. reported that co-culture of MSCs with bladder smooth muscle cells (bSMCs) potently mitigates hypoxia-induced inflammatory and profibrotic pathways, with >50% inhibition of hypoxia-induced TGFβ1 and IL-6 expression (*p* < 0.005) and increased IL-10 protein (*p* < 0.005) [29]. Moreover, Qiu et al. have demonstrated the protective effect of adipose-derived mesenchymal stem cells (AdMSCs) in the management of radiation-induced bladder dysfunction and histological changes in an in vivo model. These results suggest a potential for MSCs in the management of radiation cystitis [36]. However, their clinical application remains limited, partly due to the difficulty of isolating them and obtaining quantities sufficient for the management of these pathologies [1,8,37].

MSC-EVs and MSC-CM appear to be therapeutic candidates in tissue remodeling and repair.

EVs, including exosomes and microvesicles, are a heterogeneous population of lipid membrane-delimited nanoparticles encapsulating a plethora of bioactive factors, including proteins (cytokines, membrane receptors, growth factors and enzymes) and genetic materials (mRNAs and microRNAs) [38,39]. They function as key intercellular signaling mediators to elicit biological responses via horizontal transfer of their bioactive cargo into recipient cells [38,39,40]. A further advantage, compared with the original MSCs, is that MSC-EVs cannot self-replicate, thus allaying safety concerns associated with cell therapy, such as uncontrolled cell division and cellular contamination with tumorigenic cells [10,41]. Likewise, conditioned medium from human mesenchymal stromal cells (MSC-CM) has demonstrated therapeutic potential in various fields such as myocardial infarction, stroke and acute and chronic hindlimb ischemia, neurodegenerative diseases, spinal cord injury, alopecia, acute and chronic wounds, acute liver injury/failure, lung injury, periodontal tissue injury, male infertility, soft tissue and bone defects [21,22,23,24,25,26]. Different studies have been conducted involving the use of MSC-CM for hair follicle regeneration, fractionated carbon dioxide wound healing, and treatment of inflammatory arthritis and multiple sclerosis [42,43,44,45,46]. Thus, the MSC secretome has been suggested as a novel cell-free medicinal product that can recapitulate the beneficial effects of MSCs and has various advantages in overcoming the limitations and risks associated with cell-based therapy [40,41].

We have demonstrated that MSC-EVs and MSC-CM are candidates in the prevention of radiation cystitis with the potential to inhibit the fibrotic process, inhibit the myofibroblast phenotype, decrease the secretion of pro-fibrotic cytokines and increase the secretion of anti-fibrotics cytokines by HUBF cells, and finally through their own pro-angiogenic properties.

First of all, MSC-EVs and MSC-CM inhibit the myofibroblastic phenotype and, therefore, decrease the production of the extracellular matrix, the first key step in radiation fibrosis. Indeed, the gene expression of αSMA, CTGF and Col1α2, which play an essential role bladder fibrosis, was found to decrease.

At the same time, an increase was observed in the gene expression of MMP2. The increase in MMP2 and the decrease in Col1α2 are responsible for the degradation of fibrillar collagen [47].

MSC-EVs and MSC-CM do not influence gene expression of Col3α1, TIMP1 and TGFβ. Col3α1 and TIMP1 are involved in the wound remodeling phase and ensure homeostasis. TIMP1 is also involved in restricting both inflammation and ECM accumulation/fibrosis following injury [23]. Timp1^−/−^ mice had significantly increased injury, inflammation, and fibrosis following carbon tetrachloride-induced liver injury compared to wild type mice. These studies provide evidence of a role for TIMP1 in restricting both inflammation and ECM accumulation/fibrosis following injury [48].

Based on these results, it appears that the effect of MSC-EVs and MSC-CM on HUBFs does not shift the balance towards a pro-fibrotic phenotype but rather to a pro-healing phenotype. Indeed, the persistence of TGF-β1, TIMP1 and Col3α1 contributes to the wound-healing process following irradiation. Tutuianu et al. confirm these results in an *in vitro* wound-healing model. They reported that the regenerative properties of MSC-derived exosomes were validated using a wound-healing skin organotypic model, which exhibited full re-epithelialization upon exosomes exposure [47].

Basalova et al. demonstrated that MSC-EVs down-regulated secretion of extracellular matrix proteins by fibroblasts as well as suppressed their contractility, resulting in prevention as well as reversion of fibroblasts differentiation to myofibroblasts [49].

Moreover, MSC-EVs and MSC-CM induce a modulation of the secretome of irradiated HUBFs. Preconditioned HUBFs exhibit an anti-fibrotic secretome with inhibition of secretion of pro-fibrotic cytokines and increased secretion of anti-fibrotic proteins.

The proteins analyzed within the secretome have previously been identified as key proteins of fibrosis in clinical and preclinical studies [18,22,50,51,52]. Zwaans et al. reported a high level of HGF in patients who received a diagnosis of radiation cystitis [36]. HGF could be one key element in the prevention of RC. We demonstrated a decrease in HGF secretion from preconditioned HUBFs.

Their action is possibly linked to their high concentration of miRNAs, such as miR-21 and miR-29, which mediate antifibrotic effects [49].

Finally, vascular lesion is the second essential step in the development of radiation cystitis, starting from the acute phase. Analysis of their composition reveals a pro-angiogenic and anti-inflammatory signature. We observed an increase in the level of IL-8 protein, which stimulates endothelial cell proliferation and capillary tube organization and promotes angiogenesis by directly interacting with endothelial cells [53]. MCP-1 can also act directly on endothelial cells to induce angiogenesis [54]. Human umbilical vein endothelial cells (HUVECs) treated with MCP-1 showed significantly increased numbers of capillary-like tube formation. CXCL16 is a robust agonist of both angiogenesis (HUVEC migration and incorporation) and vasculogenesis which promotes HUVEC proliferation, chemotaxis, and tube formation by activating ERK pathway *in vitro* [55]. Moreover, we demonstrated that HUVECs from incubation with MSC-CM and with MSC-EVs formed more nodes, junctions, and longer segments than the cells incubated with medium only.

These data confirm the therapeutic potential of MSC-EVs and MSC-CM in the management of tissue repair, and in our case the prevention of radiation cystitis.

As of today, MSC-EVs and MSC-CM are being evaluated in phase I/II clinical trials for the prevention or management of severe forms of COVID-19, and also inflammatory digestive pathologies, for their anti-inflammatory properties [55,56,57,58,59,60,61]. This confirms the feasibility of these new therapeutic possibilities in clinical practice, with intravenous but also intranasal delivery.

However, with equivalent therapeutic potential, MSC-CM seems easier to use. Indeed, the techniques necessary to isolate and characterize MSC-EVs are long and can be expensive [10,62,63]. This supports MSC-CM as a potential candidate for simpler and less expensive clinical application.

## 5. Conclusions

MSC-EVs and MSC-CM appear to offer a promising therapeutic approach for management of RC and thereby an opportunity to improve patients’ the quality of life. *In vitro*, MSC-EV and MSC-CM appear to have promising therapeutic potential in the prevention of RC by targeting the three main stages of RC, fibrosis, inflammation and vascular damage.

If they have the same efficiency, then three important elements are to be emphasized on the conditioned medium of MSCs. MSC-CM is easily obtained in large quantities and can therefore be used locally by bladder instillation to locally reduce the development of radiation-induced fibrosis of the bladder. MSC-CM is richer in composition of proteins and other molecules, giving it a broader spectrum of action, or longer efficacy over time. Obtaining MSC-CM seems less expensive to obtain than MSC-EVs.

The efficacy and safety of these two approaches should be studied in a mouse model of radiation cystitis.

## Figures and Tables

**Figure 1 biology-11-00980-f001:**
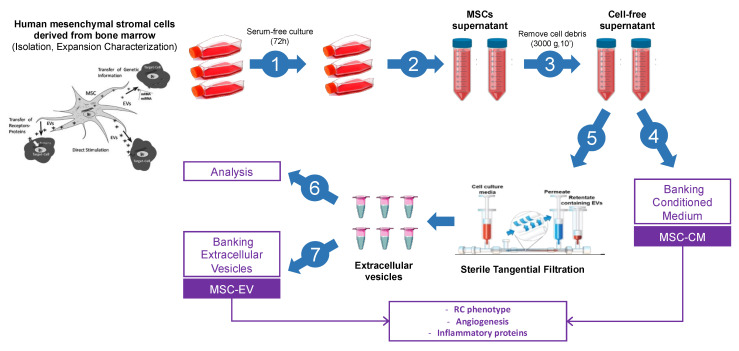
**MSC-CM and MSC-EV collection**. MSC-CM and MSC-EVs collection and characterization are performed in 7 distinct steps: (**1**) expansion, (**2**–**3**) obtain cell-free supernatants, after cell culture in serum-free medium (72 h). Cell-free supernatants divided into two parts. One part corresponding to MSC-CM (**4**). The second part filtered by tangential filtration for isolated MSCEV (**5**) and analyzed (**6**). (**7**) Banking of MSC-EV.

**Figure 2 biology-11-00980-f002:**
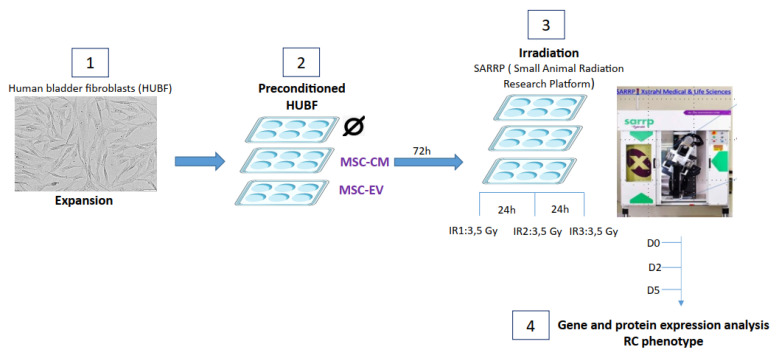
**Representation of the 4 steps for obtaining an *in vitro* model of radiation cystitis (RC):** (**1**) culture and expansion of HUBFs, (**2**) 72 h of preconditioning of HUFBs (control without conditioning (Ø)), with MSC-CM) or with (MSC-VE), (**3**) 3 irradiations of HUBFs at a dose of 3.5 Gy at 24 h intervals, using the Xstrahl small animal radiation research platform (SARRP), (**4**) harvest of supernatants and cells at 0, 2 and 5 days after last exposure (for gene and protein expression analysis).

**Figure 3 biology-11-00980-f003:**
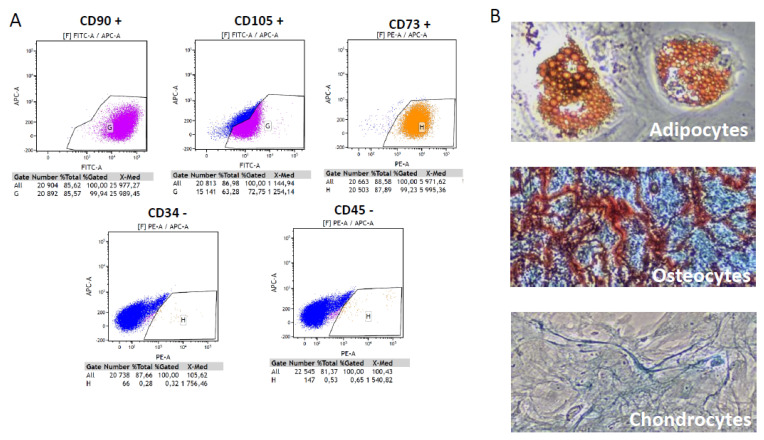
**MSC characterization**. (**A**) Flow cytometric analysis of MSC cells. The cells at passage 4 were labeled with the CD90−FITC, CD105−FITC, CD73−PE, CD34−PE and CD45−PE antibodies. The cells were positive for CD90, CD105 and CD73, and negative for the hematopoietic markers CD34 and CD45. (**B**) Assays for MSC differentiation. MSCs were cultured with lineage-specific media for 21 days to induce cell differentiation. Differentiation into adipocytes, osteocytes and chondrocytes was detected by Oil Red O, alizarin red and alcian blue, respectively (10× objective).

**Figure 4 biology-11-00980-f004:**
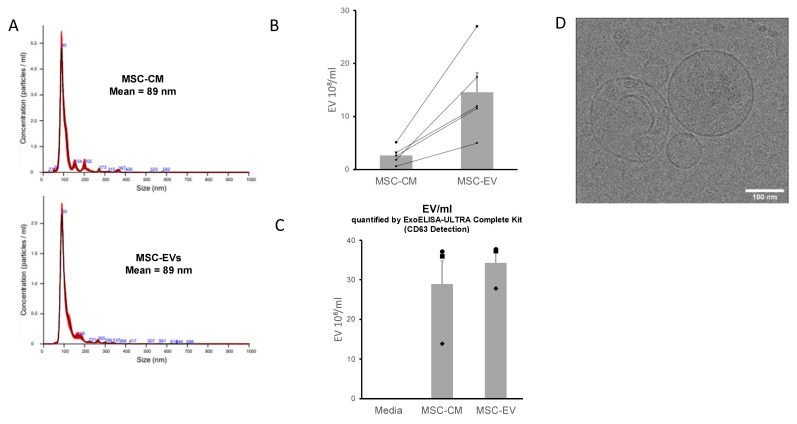
**MSC-CM and MSC-EVs characterization.** (**A**): Graph showing the calculated mean ± SD of particles size distribution of MSC-EVs and MSC-CM samples by Nanoparticles Tracking Analysis (NTA). (**B**): Quantification of particles present in MSC-EVs and MSC-CM samples by NTA, graph showing the calculated mean ± SD, *n* = 5, *p* = 0.0159. (**C**): Quantification of particles present in MSC-EVs and MSC-CM samples by ELISA specific to CD63 detection (marker of extracellular vesicles), *n* = 3. (**D**): Visualization of particles in EVs sample by cryo-electron microscopy.

**Figure 5 biology-11-00980-f005:**
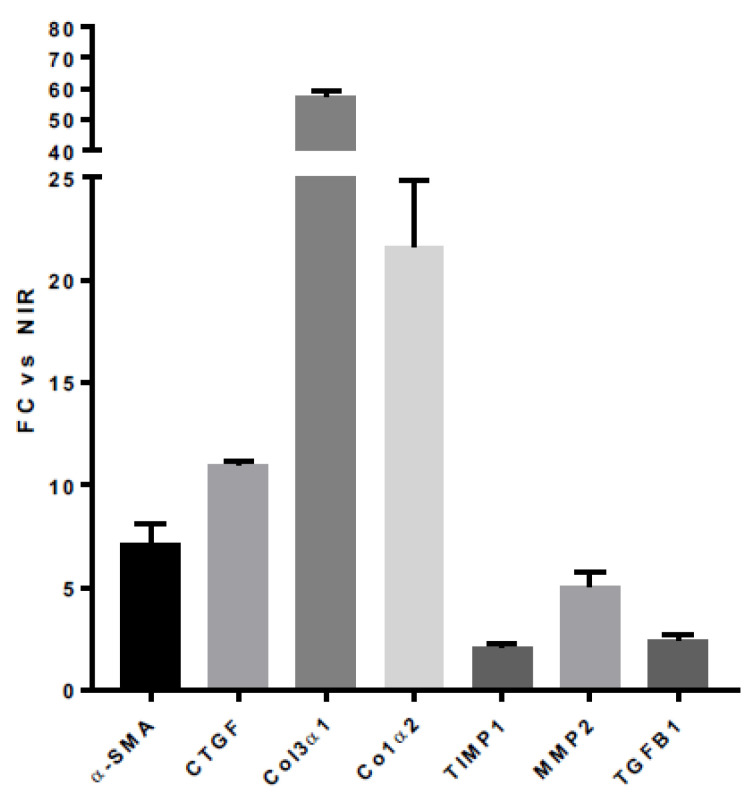
**Results of the expressions of genes involved in fibrosis in irradiated HUBFs** (3 × 3.5 Gy) at D5 post irradiation compared to non-irradiated HUBFs. The results were normalized to the untreated HUBFs. Gene expression profile of HUBFs after irradiation compared to non-irradiated control (NIR). FC: Fold change. The mRNA level of GAPDH gene was considered as normalizing control gene. Fold change values were evaluated using the following formula: 2^−ΔΔCt^, *n* = 6.

**Figure 6 biology-11-00980-f006:**
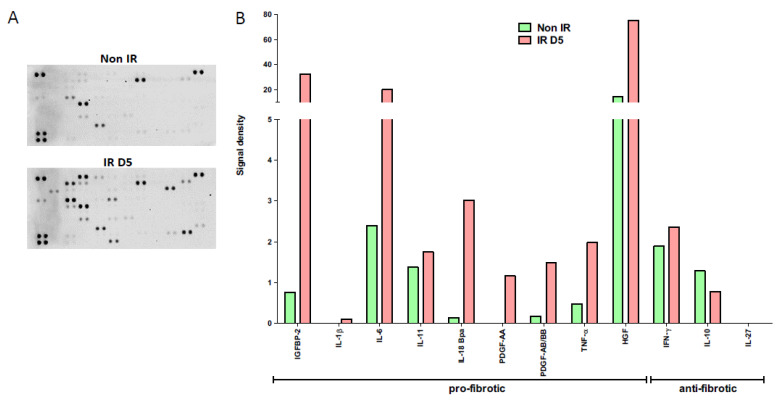
**Profiling of cytokines expressed by secretomes of irradiated or non-irradiated** fibroblasts obtained with Proteome Profiler Human XL cytokine Array Kit (R&D) (**A**) Representative image of dot blot, (**B**) graph showing signal density of dot blot for each protein normalized to positive spot pixels.

**Figure 7 biology-11-00980-f007:**
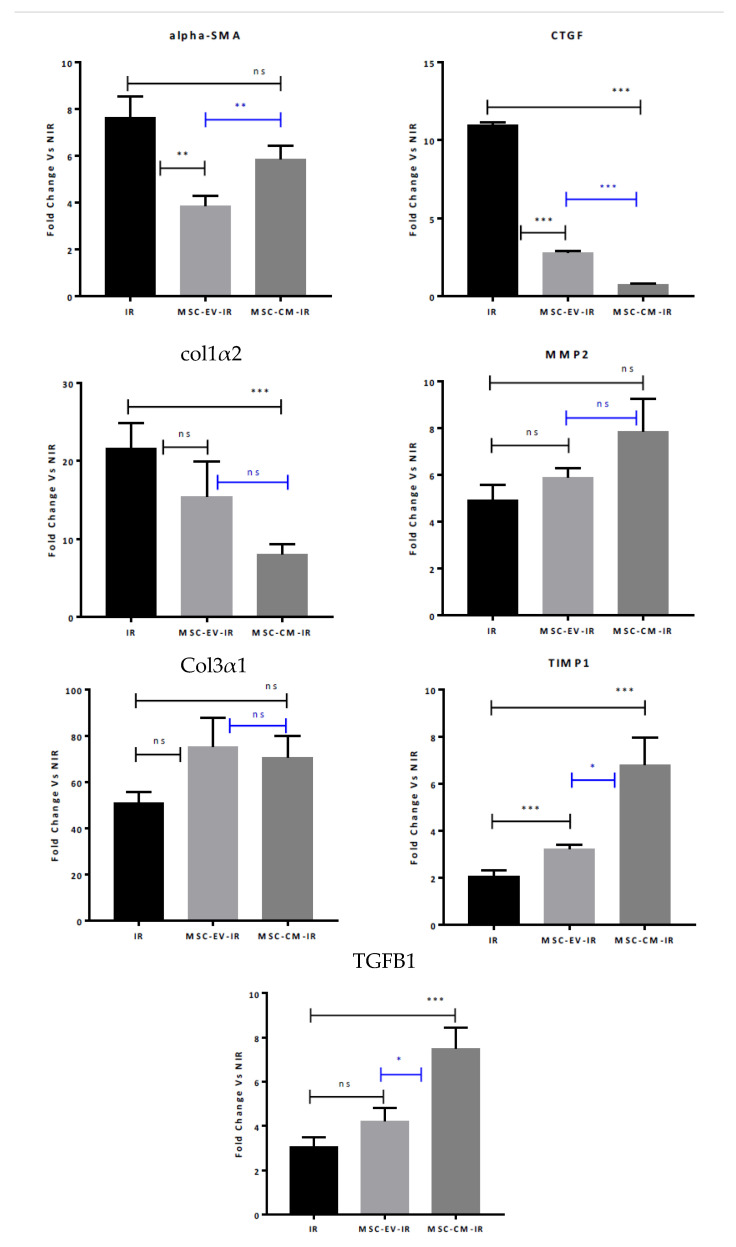
Results of gene expression of alpha-SMA, CTGF, Col1a2, MMP2, Col3α1, TIMP1 and TGFB1 in HUFBs irradiated with no treatment (IR) and HUBFs irradiated and preconditioned with MSC-EVs (MSC-EV-IR) or MSC-CM (MSC-CM-IR). Real-time PCR was used for the analysis. The mRNA level of GAPDH gene was considered as normalizing control gene. Fold change vs. NIR (Mean +/− sem), Mann–Whitney test: *p* > 0.05 ns, *p* < 0.05 *, *p* < 0.01 **, *p* < 0.001 ***, *n* = 6.

**Figure 8 biology-11-00980-f008:**
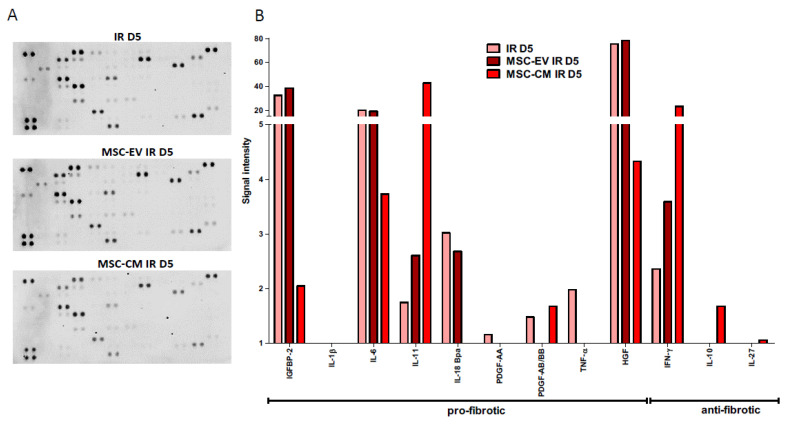
Profiling of cytokines expressed by secretomes of irradiated untreated fibroblast or preconditioned with MSC-EV or MSC-CM obtained with Proteome Profiler Human XL cytokine Array Kit (R&D). (**A**) Representative image of dot blot; (**B**) graph showing signal density of dot blot of each protein normalized to positive spot pixels.

**Figure 9 biology-11-00980-f009:**
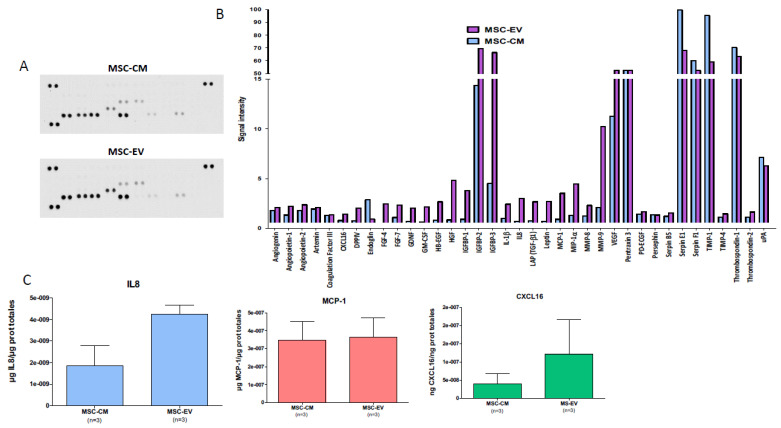
**MSC-EVs and MSC-CM protein composition**. (**A**) Profiling of angiogenesis-regulating proteins secreted by MSC-EV or MSC-CM samples obtained with Proteome Profiler Human Angiogenesis Array Kit (R&D). At the top, representative image of dot blot and at the bottom, graph showing pixel density of dot blot of each protein normalized to positive spot pixels. (**B**) Profiling of cytokines secreted by MSC-EV or MSC-CM samples obtained with Proteome Profiler Human XL cytokine Array Kit (R&D). At the top, representative image of dot blot and at the bottom, graph showing pixel density of dot blot of each protein normalized to positive spot pixels (**C**): Quantification of IL8, MCP-1 and CXCL16 secreted by MSC-EV or MSC-CM samples by ELISA (mean ± SD), n = 3.

**Figure 10 biology-11-00980-f010:**
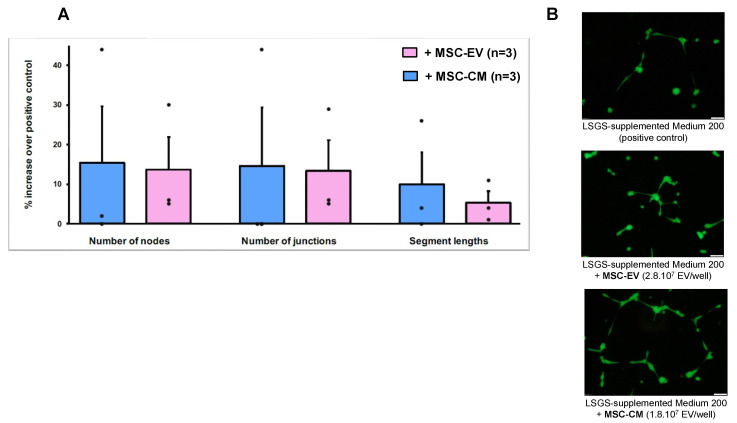
**Tube formation of HUVECs incubated with MSC-CM or MSC-EV.** (**A**) Number of nodes, number of junctions and number of segments length were quantified in different experimental conditions with the ImageJ software. The graph shows the percentage increase observed with medium + MSC-CM or + MSC-EV compared to the positive control. Values represent means ± SEM from three independent experiments. (**B**) Representative images (10× magnification, scale bar = 50 mm) of tube formation were taken using fluorescent microscopy (calcein-AM staining), 12 h after cell seeding.

## Data Availability

The data that support the findings of this study are available from the corresponding author upon request.

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
