# Peer review of "Two New Potential Therapeutic Approaches in Radiation Cystitis Derived from Mesenchymal Stem Cells: Extracellular Vesicles and Conditioned Medium"

_biology, 2022, doi:10.3390/biology11070980_

Round 1
Reviewer 1 Report
I have found that some points have been addressed but others are not.
The manuscript still presents several inaccuracy. It is still hard to read. The modlaity by which results are presented is confusing and not always clear.
Comment 1: authors have not properly reply to this comment. It is necessary to highlight the limit of the study as no damage is induced in the in vitro study so they cannot assume that the study has a therapeutic potential. So, please add this limitation.
No statistic in all figures (i.e. 4 and 5)
Figure 4 is still not properly described in the results. Please, provide. It is difficult to read
Line 296: if Ev are still present in the CM, then authors must to highlight this, because it is a limitation of the study
Figure 5 shows a more ability of CM to increase profibrotic features compared to EV: authors have not responded properly by only stating that "Our data show the potential anti-fibrotic of MSC-EVs and MSC-CM". Please, provide.
Author Response
Response to Reviewer 1 Comments
Thank you again for your consideration of our article and helpful recommendations.
We have made numerous revisions and corrections which we believe have improved the clarity and quality of our article
Comment 1: authors have not properly reply to this comment. It is necessary to highlight the limit of the study as no damage is induced in the in vitro study so they cannot assume that the study has a therapeutic potential. So, please add this limitation.
We based our in vitro model on the work of Andreassen et al (doi.org/10.1016/j.radonc.2013.08.029), Alsner et al (doi.org/10.1016/j.radonc.2007.05.001). The latter demonstrated to induce fibrosis using the irradiation schedule of 3 × 3.5 Gy with 24 h interval, as described in the Material and Methods section.
Each reported inducing a fibrosis process after irradiation of skin fibroblasts. This model has been validated. Thus, we can state from these findings that the chosen scheme induces the fibrosis process after irradiation of human bladder fibroblasts. Our gene expression results of the irradiated fibroblasts presented confirm this hypothesis, in our in vitro model. We certainly need to validate these results with an in vivo model.
No statistic in all figures (i.e. 4 and 5)
We thank the reviewer.
Figure 4 B. In the conditioned medium there is on average 2.6 +/- 0.7 x 108 EV/ml. Tangential filtration significantly increases by a factor of 5.5 (p=0.0159).
Figure 4C aims to validate the presence of extracellular vesicles (microvesicles and exosomes). The use of CD3 labeling allows to assert this presence, but does not distinguish between microvesicles and exosomes. The tangential filtration technique allows to increase the concentration of MSC-EVs and thus limit the presence of other elements such as exosomes. The reference technique to characterize MSC-EVs is NTA.
Thus this figure, n=3, allows to affirm the presence of extracellular vesicles within MSC-EVs and MSC-CMs. Thus we did not put error bars.
We thank the reviewer for this suggestion. We have done the corrections, n=3.
For figure 5: We interpreted the fold changes as follows:
If there is a doubling (fold change=2, Log2FC=1) between IR and NIR, then IR is twice as large as NIR (or IR is 200% of NIR).
If there is a two-fold decrease (factor change = 0.5, Log2FC = -1) between IR and NIR, then IR is half as large as NIR or IR is 50% of NIR.
In this study, we consider a significant overexpression from 1.5 FC and underexpression from 0.5 FC of IR versus NIR.
We have made the correction.
Figure 4 is still not properly described in the results. Please, provide. It is difficult to read
The characterization of EVs must meet the recommendations of the International Society of Extracellular Vesicles (ISEV).
In fact, to cope with ambiguity in appropriate methods, the International Society of Extracellular Vesicles (ISEV) has developed a set of criteria for the study of EVs to drive publication of accurate and reproducible data.
The Minimal Information for the Study of EVs (MISEV) include tables and outlines of suggested protocols and steps to follow to document specific EV-associated functional activities.
We have presented these different points:
- Particles size distribution by Nanoparticles Tracking Analysis (NTA)
- Quantification of particles by Nanoparticles Tracking Analysis (NTA)
- Visualization of particles in EVs sample by cryo-electron microscopy
We also performed a quantification of particles by ELISA specific to CD63 detection, as requested by the reviewers, in order to confirm the presence of extracellular vesicles (microvesicles and exosomes) in the two conditionned.
Line 296: if Ev are still present in the CM, then authors must to highlight this, because it is a limitation of the study
The beneficial effects of MSCs may be attributed to their paracrine action. This may be achieved through the conditioning of its environment by secretion of specific components (MSC-CM) and among them extracellular vesicles (MSC-EVs). MSC-CM or secretome contains soluble factors and MSC EV. MSC EV were selectively enriched in distinct class of RNAs.
We have taken into account this remark of the reviewer to better clarify the CM and Ev. Thus we have detailed figure 1 and corrected the text: The medium was collected and centrifuged (3000g,10 min) to remove whole cells and debris. A part of the supernatant was kept at -80°C (MSC-CM) and the other part was filtered by tangential flow filtration (TFF) using syringes with single-use hollow fiber filters (Repligen, mPES 500k) to isolate and concentrate MSC-EVs (Figure 1).
The EVs today are the new promising therapeutic agents. We wanted to demonstrate that the use of CM, which also contains EVs and their therapeutic potential, also contains other elements secreted by MSCs that are also useful in the process of repair and prevention of fibrosis. This is especially true since the production of CM is much easier than that of EVs, and therefore easier to apply clinically.
We have made this clarification
Figure 5 shows a more ability of CM to increase profibrotic features compared to EV: authors have not responded properly by only stating that "Our data show the potential anti-fibrotic of MSC-EVs and MSC-CM". Please, provide.
The figure 5 compares the gene expression of unirradiated HUFBs with irradiated HUFBs. And yes, we can see that irradiation induces an overexpression of genes involved in the fibrosis process. Similarly, figure 6 compares the protein expression of unirradiated HUFBs with that of irradiated HUFBs. We see that irradiation induces an increase in the secretion of pro-fibrotic proteins.
But figures 7 and 8 show that preconditioning with EV or CM leads to a decrease in most of the genes involved in the fibrosis process as well as the production of pro-fibrotic proteins. And we see an overexpression of anti-fibrotic proteins.
We specify this in the text: this semi-quantitative analysis shows a significant decrease in pro-fibrotic proteins within the secretome of irradiated HuFBs pre-conditioned with MSC-CM and MSC-EVs compared to irradiated HUBFs: IGFBP2, IL1-b, IL6, IL11, IL18, PDGF, TNFa, HGF in MSC-CM, MSC-EVs and irradiated HUBF groups respectively. Furthermore, a significant increase was observed in anti-fibrotic proteins within the secretome of irradiated HUBFs preconditioned with MSC-CM and MSC-EVs compared to irradiated HUBFs: with IFNg, IL10 and IL27 in MSC-CM, MSC-EVs and irradiated HuFB groups respectively.
We have made this clear in the text.

Reviewer 2 Report
The topic of the paper entitled "Two new potential therapeutic approaches in radiation cystitis derived from mesenchymal stem cells: Extracellular Vesicles and Conditioned Medium" is interesting. The EVs today are the new promising therapeutic agents.
In the results sub-paragraph 3.1 the authors don't explain the MSCs isolation method but only their characterization, so maybe it is better to change the title.
In subparagraph 3.2 they show the EVs and MSC-CM characterization, in the associated figure 4C how many experiments did the authors carry out? In Fig.4D they show particle visualization by cryo-electron microscopy, what about the particles in MS-CM samples?
In subparagraph 3.3, the authors demonstrate the HUBFs activation into myofibroblasts only through the analysis of gene expression, maybe a better characterization (ie immunohistochemical analysis) could improve the results.
Fig.5 FC abbreviation explanation is missing.
Fig. 6, 8, and 9 Regarding the Array assays how many experiments were carried out? Are the results representative of different biological replicates?
Fig. 8 Check the figure legend and correct typing errors.
Fig. 9 check and correct the figure legend.
Figure 10, in the tube formation assay they reported a histogram with a very high standard deviation, could they improve this result?
Author Response
Response to Reviewer 2 Comments
Thank you again for your consideration of our article and helpful recommendations.
We have made numerous revisions and corrections which we believe have improved the clarity and quality of our article
In the results sub-paragraph 3.1 the authors don't explain the MSCs isolation method but only their characterization, so maybe it is better to change the title.
We thank the reviewer, and we have made the correction.
In subparagraph 3.2 they show the EVs and MSC-CM characterization, in the associated figure 4C how many experiments did the authors carry out? In Fig.4D they show particle visualization by cryo-electron microscopy, what about the particles in MS-CM samples?
Figure 4C aims to validate the presence of extracellular vesicles (microvesicles and exosomes). The use of CD3 labeling allows to assert this presence, but does not distinguish between microvesicles and exosomes.
Thus this figure, n=3, allows to affirm the presence of extracellular vesicles within MSC-EVs and MSC-CMs. Thus we did not put error bars.
The tangential filtration technique allows to increase the concentration of MSC-EVs and thus limit the presence of other elements such as exosomes.
The reference technique to characterize MSC-EVs is NTA, according to the recommendations of the International Society of Extracellular Vesicles (ISEV) for the characterization of MSC-EVs.
Cryo-electron microscopy allows the identification of EVs after tangential filtration, when they are concentrated. In MSC-CM, EV are not concentrated so it would be quite difficult to see them in cryo-microscopy.
In subparagraph 3.3, the authors demonstrate the HUBFs activation into myofibroblasts only through the analysis of gene expression, maybe a better characterization (ie immunohistochemical analysis) could improve the results.
We thank the reviewers for this suggestion. We validated our model of radiation cystitis on pro-fibrotic gene overexpression based on the work of Andreassen et al (doi.org/10.1016/j.radonc.2013.08.029), Alsner et al (doi.org/10.1016/j.radonc.2007.05.001).
Fig.5 FC abbreviation explanation is missing.
We thank the reviewer, and we have made the correction.
Fig. 6, 8, and 9 Regarding the Array assays how many experiments were carried out? Are the results representative of different biological replicates?
We used the semi-quantitative analysis of Proteome Profiler Human XL Cytokine Array (R&D Systems Catalog Number ARY022B). The objective of this technique is to establish a cytokine profile allowing a comparison according to the conditions. Based on the recommendations of the Proteome Profiler Human XL Cytokine Array (R&D Systems Catalog Number ARY022B) and on literature data, only one experiment per condition is needed to obtain this cytokine profile and it reflects the different biological conditions. However, we have quantified by an ELISA technique the cytokines of interest, as represented in figure 9.
Fig. 8 Check the figure legend and correct typing errors.
We thank the reviewer, and we have made the correction.
Fig. 9 check and correct the figure legend.
We thank the reviewer, and we have made the correction.
Figure 10, in the tube formation assay they reported a histogram with a very high standard deviation, could they improve this result?
Whether the MSC secretome is filtered or not, it increases angiogenesis, with no significant difference on the number of nodes, junctions or lenght segments. It would be necessary to increase the number of image analysis series per condition to confirm a lack of difference in the effect of MSC-CM and MSC-VE on the different variants measured. There might be a slight difference in the lenght segments but these alone cannot represent the pro-angiogenic potential.

Reviewer 3 Report
mesenchymal stem cells: Extracellular Vesicles and Conditioned Medium” by Helissey and collaborators, the author are interested in assessing the therapeutic potential of both MS MSC-EV and MSC-EV.
The subject of the paper is really interesting and it is well organized. The author demonstrated that both MSC-EV and MSC-CM coul prevent RC by targeting fibrosis, inflammation and vascular damage. The conclusion of the authors are promising for future therapies.
Major revision
Line 255: “Expression of α-SMA, CTGF, TGF-β1, TIMP1, TIMP2, Col3α1, Col1α2 and matrix metallopeptidases (MMPs) were upregulated in irradiated HUBFs compared to non-irradiated HUBF”. In figure 5 TIMP2 is not represented, please add. Moreover, the authors evaluate the level of TIMP1 but not that of MMP9 of which TIMP1 is the inhibitor. Please add the experiment.
In section 3.5 the authors studied the expression of some pro-fibrotic/anti-fibrotic genes after irradiation. In my opinion to this experiment they have to add also TIMP2 as it is supposed that they have already taken it into account (section 3.3) and because it is reported in the literature that it has a fibrotic role (e.g., Valerie Arpino, et al. The role of TIMPs in regulation of extracellular matrix proteolysis. Matrix Biology Volumes 44–46 2015 Pages 247-254). Please also add MMP9 which is involved in fibrosis regulation too (Wang Q, et al. Mol Med Rep. 2019 Dec;20(6):5239-5248. doi:10.3892/mmr.2019.10740. Mohamed HA, Said RS. Int Immunopharmacol. 2021 Mar;92:107347. doi:10.1016/j.intimp.2020.107347)
Minor revision
Lines 152-153: “LSGS-supplemented Medium 200 with MSC-EV (2.5.107VE/well) or LSGS-supplemented Medium 200 with MSC-CM (1.8.107VE/well).” It is supposed to be 2.5 x 107 EV/well and 1.8 x 107 EV/well. Please correct.
Line 239: “TIMP2, Col3α1, Col1α2 and matrix metallopeptidases (MMPs) were upregulated in irradiated HUBFs compared to non-irradiated HUBF. Please correct the exponential numbers.
Line 482: “inflammatory arthritis (26), and multiple sclerosis (27).” Reference 27 does not speak about MS please correct and check all the other references.
Line 496: extra space.
Please write CO2 correctly all over the paper.
Author Response
Thank you again for your consideration of our article and helpful recommendations.
We have made numerous revisions and corrections which we believe have improved the clarity and quality of our article
Line 255: “Expression of α-SMA, CTGF, TGF-β1, TIMP1, TIMP2, Col3α1, Col1α2 and matrix metallopeptidases (MMPs) were upregulated in irradiated HUBFs compared to non-irradiated HUBF”. In figure 5 TIMP2 is not represented, please add. Moreover, the authors evaluate the level of TIMP1 but
We evaluated TIMP2 and MMP9 in both conditions in irradiated HUBFs compared to non-irradiated HUBF. There was no difference in gene expression under these conditions. Therefore, the results are not shown. We make the correction in the sentence for TIMP2.
In section 3.5 the authors studied the expression of some pro-fibrotic/anti-fibrotic genes after irradiation. In my opinion to this experiment they have to add also TIMP2 as it is supposed that they have already taken it into account (section 3.3) and because it is reported in the literature that it has a fibrotic role (e.g., Valerie Arpino, et al. The role of TIMPs in regulation of extracellular matrix proteolysis. Matrix Biology Volumes 44–46 2015 Pages 247-254). Please also add MMP9 which is involved in fibrosis regulation too (Wang Q, et al. Mol Med Rep. 2019 Dec;20(6):5239-5248. doi:10.3892/mmr.2019.10740. Mohamed HA, Said RS. Int Immunopharmacol. 2021 Mar;92:107347. doi:10.1016/j.intimp.2020.107347)
We agree with others that in other models these proteins have a major role. But in our in vitro radiation cystitis model, we did not find this involvement.
Minor revision
Lines 152-153: “LSGS-supplemented Medium 200 with MSC-EV (2.5.107VE/well) or LSGS-supplemented Medium 200 with MSC-CM (1.8.107VE/well).” It is supposed to be 2.5 x 107EV/well and 1.8 x 107 EV/well. Please correct.
We thank the reviewer, and we have made the correction.
Line 239: “TIMP2, Col3α1, Col1α2 and matrix metallopeptidases (MMPs) were upregulated in irradiated HUBFs compared to non-irradiated HUBF. Please correct the exponential numbers.
We thank the reviewer, and we have made the correction.
Line 482: “inflammatory arthritis (26), and multiple sclerosis (27).” Reference 27 does not speak about MS please correct and check all the other references.
We thank the reviewer, and we have made the correction.
Line 496: extra space.
We thank the reviewer, and we have made the correction.
Please write CO2 correctly all over the paper.
We thank the reviewer, and we have made the correction.

Reviewer 4 Report
In this article, the authors report the protective effects of mesenchymal stem cell-derived extracellular vesicles (MSC-EVs) and secretome (conditioned media) against the radiation induced cystitis. Authors show that EVs and conditioned media from MSCs when transferred to irradiated human bladder fibroblasts, demonstrated the protective effects against radiation cystitis in vitro, accompanied by down-regulation of α-SMA and CTGF transcription. The elevated levels of anti-fibrotic cytokine release and decrease in the secretion of pro- fibrotic cytokines was observed. The EVs or conditioned media from MSCs induced the vessel formation in HUVEC cells.
The current study is an interesting piece of work, the study design is technically sound, and conclusions are supported with results. While the manuscript is a potential candidate for publications, it can be revised before production.
My comments and suggestions are appended below.
1. In the introduction, please provide more literature, briefly about radiation therapy in general, then radiation cystitis specifically, and limitations, associated risks, available treatments and the need for alternative treatment options.
2. While authors introduce effects of radiation on cells or organs, include a literature where conditioned media from irradiated cells have been transferred to other cells, and bystander effects of CM were reported. I suggest authors to cite the following article (PMID: 31842899).
3. Methods: the duration of angiogenesis assay (after EV or CM treatment) was 12h. The tube formation/angiogenesis takes several days, and HUVEC cell are not the exception, i.e., they form network over days, but not in hours. Please check if experimental conditions are reported properly.
4. Please mention in the text what was the HUVEC culture time before treating them with EVs/CM.
5. Regarding the induction of 3×3.5Gy, why authors did not consider exposing cells to higher doses of radiation (Gy) to HUBF, and then test the protection effects of MS-EVs and MSC-CM?.
6. Page 4: In the methods perhaps better to separate RNA extraction and reverse transcription section from in vitro radiation model, and describe it with qPCR.
7. Figure 4C: no error bars in fig 4C? better to mention the number of replicates in legends (n =), in this case.
8. Figure 5: either write the results were normalized to the untreated HUBFs. Or Gene expression profile of HUBFs after irradiation compared to non-irradiated control (NIR:). The word compare should refer to compared to non-irradiated control?. Comparison, and normalization can confuse the reader. Same for the Y-axis of the scale bar (the word vs does not show which panel is treated and which one is untreated). Either show the comparison of treated and untreated in compared barographs or simply write normalized to. Also define FS in the legends.
9. In figure legends please mention the number of replicates (n =). The panels showing bar graphs, especially where statistical analysis is applied, please replace the bar charts by dot plots to show the position/distribution of individual biological replicates. Please see Weissgerber TL, Milic NM, Winham SJ, Garovic VD. (2015) Beyond Bar and Line Graphs: Time for a New Data Presentation Paradigm. PLoS Biol 13(4): e1002128
Others
i. lung, kidney, bone marrow or cartilage... (6,7). Instead of doted lines, perhaps it can be written ‘’among others’’. lung, kidney, bone marrow or cartilage, among others (6,7).
ii. MSC-EVs et MSC-CM collection → isolation of MSC-EVs and MSC-CM collection
Author Response
Thank you again for your consideration of our article and helpful recommendations.
We have made numerous revisions and corrections which we believe have improved the clarity and quality of our article
In this article, the authors report the protective effects of mesenchymal stem cell-derived extracellular vesicles (MSC-EVs) and secretome (conditioned media) against the radiation induced cystitis. Authors show that EVs and conditioned media from MSCs when transferred to irradiated human bladder fibroblasts, demonstrated the protective effects against radiation cystitisin vitro, accompanied by down-regulation of α-SMA and CTGF transcription. The elevated levels of anti-fibrotic cytokine release and decrease in the secretion of pro- fibrotic cytokines was observed. The EVs or conditioned media from MSCs induced the vessel formation in HUVEC cells.
The current study is an interesting piece of work, the study design is technically sound, and conclusions are supported with results. While the manuscript is a potential candidate for publications, it can be revised before production.
My comments and suggestions are appended below.
- In the introduction, please provide more literature, briefly about radiation therapy in general, then radiation cystitis specifically, and limitations, associated risks, available treatments and the need for alternative treatment options.
As suggested, we have made a number of modifications to improve our introduction. This includes the following excerpts:
External pelvic radiation therapy is an important tool in the treatment modality for pelvic cancers such as prostate cancer. Irradiation techniques have improved over time, such as Intensity Modulated Radiation Therapy (IMRT), stereotactic radiation therapy, and image-guided brachytherapy. These techniques allow to deliver increasingly effective doses in smaller volumes while significantly improving treatment tolerance (1).
However, the bladder is a critical organ sensitive to low-dose radiation. Despite improvements in technology, pelvic irradiation remains a cause of acute and/or late adverse events affecting the bladder. The term "radiation cystitis" includes all damage and symptoms of the bladder following radiation to the pelvic organs.
Severity is related to radiation exposure, total dose administered, and schedule of administration and fractionation. Patients with hypertension, diabetes mellitus, history of abdominal surgery, and concurrent chemotherapy have been reported to have a higher risk of radiation cystitis, especially in advanced patients (2). Manea et al suggested that bladder neck, after high dose exposures (such as after brachytherapy treatment) may be at higher risk of (3).
Acute radiation cystitis was defined as any adverse event that occurred during or within three months of the end of radiation therapy. Clinical symptoms may include pollakiuria, cystalgia, increased frequency and frequency of urination during the day and night (polyuria), dysuria, and increased urinary urgency. Late radiation cystitis is defined as a pelvic radiation-related adverse event that occurs at least three months and possibly years after completion of radiation therapy. The most typical clinical feature is repeated hematuria of varying severity.
These adverse events can affect the patient's quality of life. The clinical management of storage symptoms for acute and late radiation cystitis is largely symptomatic with analgesics, anti-inflammatory drugs, Intravesical Instillations, Hyperbaric Oxygen Therapy (HBOT) et pouvant aller jusqu’à Cystectomy and Urinary Diversion. Good hydration is recommended for patients in order to increase diuresis, cleanse the bladder, and avoid urinary obstruction resulting from blood clots (4).
- While authors introduce effects of radiation on cells or organs, include a literature where conditioned media from irradiated cells have been transferred to other cells, and bystander effects of CM were reported. I suggest authors to cite the following article (PMID: 31842899).
We thank the reviewer for this suggestion. We have included this recommendation in our article:
This modification of the secretome of irradiated HUBF on line the bystander effects of ionizing radiation (the non-targeting effects irradiation), and thus promotes the fibrosis process. This bystander effect has been reported in other in vitro models, notably in the process of angiogenesis (26).
- Methods: the duration of angiogenesis assay (after EV or CM treatment) was 12h. The tube formation/angiogenesis takes several days, and HUVEC cell are not the exception, i.e., they form network over days, but not in hours. Please check if experimental conditions are reported properly.
According to Angiogenesis Starter Kit, Gibco, protocol, HUVECs usually form well-developed tube networks after 14–18 hours under these conditions. After 24 hours, endothelial cells typically undergo apoptosis. We followed this referenced protocol for our different experiments.
- Please mention in the text what was the HUVEC culture time before treating them with EVs/CM.
We thank the reviewer for this suggestion. We have done the corrections.
Human umbilical vein endothelial cells (HUVEC, Gibco) were removed from culture, after 7days, in LSGS-supplemented Medium 200 (Gibco), trypsinized and resuspended in LSGS-supplemented Medium 200.
- Regarding the induction of 3×3.5Gy, why authors did not consider exposing cells to higher doses of radiation (Gy) to HUBF, and then test the protection effects of MS-EVs and MSC-CM?.
We based our in vitro model on the work of Andreassen et al (doi.org/10.1016/j.radonc.2013.08.029), Alsner et al (doi.org/10.1016/j.radonc.2007.05.001). The latter demonstrated to induce fibrosis using the irradiation schedule of 3 × 3.5 Gy with 24 h interval, as described in the Material and Methods section.
Each reported inducing a fibrosis process after irradiation of skin fibroblasts. This model has been validated. Thus, we can state from these findings that the chosen scheme induces the fibrosis process after irradiation of human bladder fibroblasts and assess the protection effects of MSC-EVs and MSC-CM. Our gene expression results of the irradiated fibroblasts presented confirm this hypothesis, in our in vitro model. We certainly need to validate these results with an in vivo model.
- Page 4: In the methods perhaps better to separate RNA extraction and reverse transcription section from in vitro radiation model, and describe it with qPCR.
We thank the reviewer for this suggestion. We have done the corrections.
- Figure 4C: no error bars in fig 4C? better to mention the number of replicates in legends (n =), in this case.
Figure 4C aims to validate the presence of extracellular vesicles (microvesicles and exosomes). The use of CD3 labeling allows to assert this presence, but does not distinguish between microvesicles and exosomes. The tangential filtration technique allows to increase the concentration of MSC-EVs and thus limit the presence of other elements such as exosomes. The reference technique to characterize MSC-EVs is NTA.
Thus this figure, n=3, allows to affirm the presence of extracellular vesicles within MSC-EVs and MSC-CMs. Thus we did not put error bars.
We thank the reviewer for this suggestion. We have done the corrections, n=3.
- Figure 5: either write the results were normalized to the untreated HUBFs. Or Gene expression profile of HUBFs after irradiation compared to non-irradiated control (NIR:). The word compare should refer to compared to non-irradiated control?. Comparison, and normalization can confuse the reader. Same for the Y-axis of the scale bar (the word vs does not show which panel is treated and which one is untreated). Either show the comparison of treated and untreated in compared barographs or simply write normalized to. Also define FS in the legends.
The mRNA level of GAPDH gene was considered as normalizing control gene. Data from Real-time PCR analysis of mRNA expression for alpha-SMA, CTGF, Col3a1, cola1a2, TMIP1, MMP2 and TGFB1, 5 days after radiation exposure were normalized against GAPDH expression. DDCT= DCt irradiated HUBFs – DCt non irradiated (NIR) HUBFs. Fold change (FC) values were evaluated using the following formula: 2 −ΔΔCt noted on the Y axis as FC vs NIR.
Six biologically independent experiments were done.
- In figure legends please mention the number of replicates (n =). The panels showing bar graphs, especially where statistical analysis is applied, please replace the bar charts by dot plots to show the position/distribution of individual biological replicates. Please see Weissgerber TL, Milic NM, Winham SJ, Garovic VD. (2015) Beyond Bar and Line Graphs: Time for a New Data Presentation Paradigm. PLoS Biol 13(4): e1002128
We thank the reviewer for this suggestion. We have done the corrections
Others
- lung, kidney, bone marrow or cartilage... (6,7). Instead of doted lines, perhaps it can be written ‘’among others’’. lung, kidney, bone marrow or cartilage, among others (6,7).
We thank the reviewer for this suggestion. We have done the corrections
- MSC-EVs et MSC-CM collection → isolation of MSC-EVs and MSC-CM collection
We thank the reviewer for this suggestion. We have done the corrections

Round 2
Reviewer 1 Report
This Reviewer has been satisfied.
Author Response
We thank the reviewers.
Reviewer 3 Report
Please see the attachment.

Author Response
Response to Reviewer 3 Comments (round 2)
Thank you again for your consideration of our article and helpful recommendations.
We have made numerous revisions and corrections which we believe have improved the clarity and quality of our article.
Major revision
However, there is still something which is not clear. In the response to reviewer 3 comments
the authors the authors answering to reviewer comment referred to line 255 (original version)
write: “We evaluated TIMP2 and MMP9 in both conditions in irradiated HUBFs
compared to non-irradiated HUBF. There was no difference in gene expression under
these conditions. Therefore, the results are not shown. We make the correction in the
sentence for TIMP2.”
This correction it is nont evident, so please better elucidate it.
In addition, the reviewer, in accordance to literature data, suggested the authors to sudy both
MMP9 and TIMP2.
Their answer was: “We agree with others that in other models these proteins have a
major role. But in our in vitro radiation cystitis model, we did not find this involvement.”
Whether Helissey et al., have investigated MMP9 and TIMP2 pathway and didn’t find
any involvement, they have to discuss this result within the paper.
Otherwise, due to their well known role in fibrosis they have to add experiments to investigate
their involvement in authors’ RC model.
We agree with the reviewer that TIMP 2 and MMP 9 have an important role in the fibrosis process, as in the liver fibrosis (Wang et al, Mol Med Rep 2019) and digestive fibrosis models (Mohamed et al. Int Immunopharmacol. 2021). However, the role of TIMP2 and MMP9 remains to be defined in the radiation cystitis model. We evaluated TIMP2 and MMP9 in both conditions, in irradiated HUBFs compared to non-irradiated HUBF. There was no difference in gene expression under these conditions. Similarly, Zwaans et al. did not reveal any difference in TIMP2 secretion in prostate cancer patients with symptoms of radiation cystitis compared to prostate cancer patients who were irradiated but without symptoms of radiation cystitis and compared to the control population without prostate cancer.
The place of these molecules in RC still needs to be explored.
We have discussed this point in our revised article.
Other molecules are also involved in the fibrosis process such as MMP9 and TIMP2. Indeed, Mohammed and Said reported MMP-9 is constitutively expressed in the intestinal mucosa, and is highly expressed in rat intestines after ϒ-radiation exposure, and its inhibition leads to a tendency to inhibit fibrosis. Wang et al confirm the role of MMP9 in the process of liver fibrosis, but its role is dynamic over time. And the different properties of MMP9 must continue to be explored.
In our model, we did not find any significant evolution of MMP9 and TIMP2 gene expression between irradiated and non-irradiated HUBF. Zwaans et al report no difference in PAI, TIMP1 and TIMP2 expression in urine between prostate cancer patients who underwent pelvic irradiation. Similarly, there was no significant difference in the expression of PAI, TIMP1 and TIMP2 in long-surviving patients with CR symptoms compared to patients who also received pelvic radiotherapy but did not show CR symptoms.
Thus, we need further investigation to determine the place of these molecules in the radiation cystitis model.
Minor revision
Even if the authors corrected all the text, there are still to many editing erros.
We thank the reviewer, and we have made the corrections.
Line 26 in vitro →in vitro
Line 36 3 x 3.5 Gy and line 313 3*3.5 Gy
Line 121 0,16.106 → 0.16 x 106 Write in the same way all over the paper.
Line 182 2.3.1. In Vitro Tube Formation Assay → In Vitro Tube Formation Assay
Line 196 same as above
Line 189 2.5.107VE/well → 2.5 x 107 VE/well
Line 190 same as above
Line “227 TaqMan gene expression assays gene expression assays” is this subtitle correct?
Line 269- CD 45+, CD 34+→ CD 45+
Line 274-278 please justify
Line 275 an/bodies → antibodies
Line 285 1,45 x 109 →1.45 x 109
Line 304 Col1 2→Col1α2
Line 514 Qiu et al→ Qiu et al.

Reviewer 4 Report
The authors have addressed my comments and made adequate improvements.
I endorse the publication of this work.
Author Response
We thank the reviewer.
Round 3
Reviewer 3 Report
The authors answered to all the reviewer request. There are still several editing mistakes to be corrected.
Line 121: 0,16 x 109 → 0.16 x 109
Line 155: 1.1.109 → 1.4 x 109
Line 187: 5.104 → 5 x 104
Line 190: 1.8.107→ 1.8 x 107
Line 203: et val 2013→ et al. 2013
Line 209: 1.45.108→ 1.45 x 108
Line 227: TaqMan gene expression assays gene expression assays→ TaqMan gene expression assays
Line 411: cytokine Array Kit (R&D).( A): → cytokine Array Kit (R&D). (A):
Legend figure 10: please use the same font of all the paper
Author Response
Thank you again for your consideration of our article and helpful recommendations.
We have made numerous revisions and corrections which we believe have improved the clarity and quality of our article.
The authors answered to all the reviewer request. There are still several editing mistakes to be corrected.
We thank the reviewer, and we have made the corrections.
Line 121: 0,16 x 109 → 0.16 x 109
Line 155: 1.1.109 → 1.4 x 109
Line 187: 5.104 → 5 x 104
Line 190: 1.8.107→ 1.8 x 107
Line 203: et val 2013→ et al. 2013
Line 209: 1.45.108→ 1.45 x 108
Line 227: TaqMan gene expression assays gene expression assays→ TaqMan gene expression assays
Line 411: cytokine Array Kit (R&D).( A): → cytokine Array Kit (R&D). (A):
Legend figure 10: please use the same font of all the paper
